# Health Impact and Therapeutic Manipulation of the Gut Microbiome

**DOI:** 10.3390/ht9030017

**Published:** 2020-07-29

**Authors:** Eric Banan-Mwine Daliri, Fred Kwame Ofosu, Ramachandran Chelliah, Byong Hoon Lee, Deog-Hwan Oh

**Affiliations:** 1Department of Food Science and Biotechnology, Kangwon National University, Chuncheon 200-701, Korea; ericdaliri@yahoo.com (E.B.-M.D.); ofosufk17@kangwon.ac.kr (F.K.O.); ramachandran865@gmail.com (R.C.); deoghwa@kangwon.ac.kr (D.-H.O.); 2Department of Microbiology/Immunology, McGill University, Montreal, QC H3A 2B4, Canada; 3SportBiomics, Sacramento Inc., Sacramento, CA 95660, USA

**Keywords:** microbiome, biomarkers, personalized nutrition, microbes, metagenomics

## Abstract

Recent advances in microbiome studies have revealed much information about how the gut virome, mycobiome, and gut bacteria influence health and disease. Over the years, many studies have reported associations between the gut microflora under different pathological conditions. However, information about the role of gut metabolites and the mechanisms by which the gut microbiota affect health and disease does not provide enough evidence. Recent advances in next-generation sequencing and metabolomics coupled with large, randomized clinical trials are helping scientists to understand whether gut dysbiosis precedes pathology or gut dysbiosis is secondary to pathology. In this review, we discuss our current knowledge on the impact of gut bacteria, virome, and mycobiome interactions with the host and how they could be manipulated to promote health.

## 1. Introduction

The human body consists of mammalian cells and many microbial cells (bacteria, viruses, and fungi) which co-exist symbiotically. Recent studies have estimated the total number of human cells to be about 3–4 × 10^13^ [1,2]; yet the number of bacterial cells in the gut remains debatable. A recent study, however, estimated the total number of bacterial cells in the gut to be about 3.8 × 10^13^ [2], which implies that the ratio of human to bacteria cell ratio is almost a 1:1 ratio. This is contrary to the earlier and popular report that estimated the human cell to gut bacteria cell ratio to be 1:10 [3,4,5]. Despite the controversy, it is widely accepted that gut bacteria play essential roles in host physiology [6,7]. The gut virome (the set of all endogenous retroviruses, eukaryotic viruses, and bacteriophages in the gut) has been recognized as an important part of the gut microbiome in many studies [8,9]. They significantly outnumber the population of gut bacteria and engage in complex relationships with the host and other members of the gut microbiome [10]. Current knowledge of the strong association between gut viruses and health and chronic diseases such as cancer, diabetes, and inflammatory bowel disease (IBD) [9,11] is making the gut virome an important part of the gut microbiome study. Moreover, mutualistic interactions between viruses and bacteria, viruses and fungi [12], viruses and host, and other viral interactions with multiple players of the holobiont are being discovered [13] that will increase our understanding of the role of the gut microbiome. Gut viruses have been shown to enhance bacterial resistance to antibiotics, acid, and osmotic and oxidative stresses [14]. Another member of the gut microbiome is the mycobiome, which consists of the diverse array of fungal communities largely dominated by the phyla Ascomycota, Basidiomycota, and Zygomycota [15,16,17]. Several studies have suggested that the gut mycobiota is altered during disease [18,19] and that they play critical roles in maintaining gut homeostasis and controlling systemic immunity [20,21]. It is now widely accepted that bacteria–fungi and host–fungi interactions are essential for host health [22,23]. A deeper understanding of microbiome–host molecular interactions will be important for developing effective strategies to rehabilitate perturbed gut microbial ecosystems as a means to restore health and/or prevent disease. Though there is evidence that the interactions between members of the gut microbiome and their host cells have a significant impact on host physiology, the microbes display niche specificity [24]. Furthermore, the number and types of microbes in individuals displaying similar phenotypes and genotypes may vary significantly [25]. For this reason, it is challenging to define what a healthy gut microbiome is.

## 2. Niche Specificity of the Gut Microbiota

Gut microbes display niche specificity as distinct fungi [24] and bacteria communities (quality and quantities) exist in different compartments of the gut [26,27]. As shown in Figure 1, this is particularly true because factors such as pH [28], available substrates or nutrients [29], and the mucus structure in gut compartments [30], which may affect the type of bacteria present in distinct sites of the gut at any given time. In humans, fungal [23,31] and bacterial niches are established at different body sites by 6 weeks of age and this implies that certain factors affect the establishment of the microbiota in early life [32]. Although the source of the pioneer microbiota remains unclear as many early studies suggest ex utero colonization [33,34] while other studies propose an in utero colonization hypothesis [35,36], it is evident that factors such as local nutrient utilization and production of biomolecules that control interspecies competition and colonization strongly influence early-life microbiome assembly [37,38]. To demonstrate how interspecies cooperativity enhances colonization resistance against pathogens in the gut, Caballero et al. orally administered a bacteria cocktail (*Parabacteroides distasonis*, *Bacteroides sartorii*, *Clostridium bolteae*, and *Blautia producta*) to mice and observed that ingested bacteria effectively prevented colonization by vancomycin-resistant *Enterococcus faecium* [39].

In each niche, microbes produce bacteriocins and quorum-sensing molecules for selective inclusion or exclusion and modification of the activities of other species in the local environment [40,41]. For instance, *Pseudomonas aeruginosa* controls the proliferation of *Candida albicans* and *C. albicans* biofilm formation by secreting signaling molecules such as pyocyanin and phenazines into their environment [42,43]. The SPbeta group of phages has been shown to produce signal-hexapeptide molecules in their bacteria host cells, which guide progeny phages making lysis and lysogeny decisions [44]. It has also been found recently that *Aeromonas* phage ΦARM81ld responds to signaling molecules produced by Gram-negative bacteria [45]. To competitively exclude other species from their niche, microbes may regulate host immunity as a means of ensuring selective pressure on microbes in a niche [46].

## 3. Variability of the Gut Microbiome

Generally, the gut microbes at different sites of the gut are fairly conserved at the phyla level but high interindividual variation exists at the species and strain levels [47]. One of the most studied determinants of the variation is host genetics [25,48], but the correlation between host genotype and the microbiota in humans seems to show weak influences. It has been shown that monozygous twins may have different gut viromes if they are living in different environments/households, whereas family members living in the same household/environment have similar gut viruses, even if they are not related [49]. Meanwhile, though such twins may share some core gut mycobiomes, a high degree of inter- and intrapersonal variability exists in their fungal communities [50]. Conversely, recent studies have shown great similarity in the gut bacteria of members of human families [51,52,53], but this similarity could be confounded by common environmental exposures [54,55]. For instance, significant functional differences were found in the gut bacteria of monozygous twins exposed to different environments [56] and those with different immune statuses [57,58]. Similarly, results from a study involving 1046 healthy subjects have shown that the environment plays a stronger role in shaping the gut microbiota than genetic factors [59]. This implies that a combination of host genetics and environmental exposure (including diet) contributes in shaping the gut microbiota. The impact of diet on gut microbiota composition and function is well established [60,61,62]. Other important factors that account for the variations in the gut microbiota and their functions include drugs [63], antibiotic use [64], sex hormones [65], toxicants [66], age [67], and exercise [68,69]. Taken together, all these factors shape the gut microbiome of an individual resulting in a distinct and personalized microbial fingerprint (Figure 2).

## 4. Role of the Gut Microbiota in Immunity and Homeostasis

The human gut microbiome possesses over 9.9 million functionally distinct microbial genes [70] and has evolved to co-exist in a mutualistic relationship with the gut [6]. They influence host endocrine functions [65], production of short-chain fatty acids [71], vitamins [72], anti-inflammatory compounds [73], anti-proliferative molecules [74], and many other biomolecules, whose functions remain unknown. Indeed, the intestinal immune system is the most important line of defense against enteric microorganism invasion. The gut microbiota plays a key role in the development and maintenance of local and systemic innate [75] and adaptive immune functions [76]. In healthy people, their balanced gut microbiota composition is in harmony with the mucosal immune system. However, when the gut microbiota is severely perturbed, the mucosal immune response is usually triggered [77], thus making gut-derived diseases possible. Gut-derived diseases such as inflammatory bowel disease, obesity, and type 2-diabetes are frequently associated with a reduction in microbial diversity [78] and depletion of certain specific bacteria, including *Faecalibacterium* and *Akkermansia*, which are perceived to promote immune tolerance [79,80,81]. It has been shown that *Saccharomyces* [82] and *Bacteroides ovatus* [83] induce IgA production (an antibody necessary for neutralizing invasive pathogens in the gut) while *Lactobacillus reuteri* is required for inducing CD4^+^CD8αα^+^ double-positive intra-epithelial lymphocytes in germ-free mice [84]. It is also widely accepted that the genus *Clostridium* induces the accumulation of regulatory T cells [85], which are required for sustaining immune tolerance to allergens [86]. Moreover, *Clostridium* clusters IV, XIVa, and XVIII induce CD4^+^Foxp3^+^ T reg cells using the short-chain fatty acids they produce [87,88,89]. The cell surface of *Bacteroides fragilis* possesses surface polysaccharide A, which binds to Toll-like receptor 2 on dendritic cells to induce IL-10 by regulatory T cell production [90,91] and enhance immune tolerance [92]. The immunomodulatory ability of yeast is known to be due to presence of β-glycan on their cell walls, which binds to intestinal dendritic cells to induce primarily Foxp3^+^, IL-10^+^, and IL-17^+^ T cells [93]. Unlike the case for bacteria, relatively little work has been carried out to determine causality between the gut virome as well as mycobiome and the immune system. Current knowledge on the immunoprotective effects of the gut mycobiota is based on the studies that used antifungal agents to perturb gut fungi and assessed the effects on host physiology [94]. In fact, most gut fungal studies mainly describe associations or correlations between changes in the gut mycobiome and physiology [95,96,97]. Thus, evidences from human and murine studies show that the gut microbiome is important for promoting proper physiological and immune development. Current understanding on adaptive and innate immunity by gut microbiota and metabolites may open up new avenues for prevention and/or mitigating inflammatory conditions [46].

## 5. Microbiome Perturbations, Immune Dysfunction, and Chronic Disease

### 5.1. Obesity and Type-2 Diabetes Mellitus (T2DM)

Obesity and T2DM are interlinked [98,99] and have been found to be strongly associated with gut microbial dysbiosis [100,101]. However, it is not yet established whether the changes in certain bacterial populations could be strong biomarkers of obesity or T2DM. While the Firmicutes to *Bacteroides* ratio in the gut may be higher during obesity but reduced in lean people [102,103,104,105], a systematic review of available literature indicates that the differences do not represent a consistent feature for distinguishing lean individuals from obese ones [106]. Interestingly, some human studies showed no changes in the *Bacteroidetes*/Firmicutes ratio between fecal samples of lean and obese subjects [107,108,109,110]. However, any studies have found associations between gut bacteria and “hunger hormones” (i.e., leptin and ghrelin) [111]. More specifically, it has been shown that fat-inducing gut bacteria may repress the expression of fat-suppressing neuropeptide genes *Gcg* and *Bdnf* to promote fat accumulation [112]. Regarding gut mycobiota studies, Rodríguez et al. [17] observed high populations of *Mucor racemosus,*
*Mucor fuscus*, and *Nakaseomyces* in the gut of non-obese subjects relative to obese subjects. The fungal populations were however restored after diet-induced weight loss. An earlier study showed that transplanting fecal microbiota from lean, healthy people to patients with metabolic syndrome significantly improved insulin sensitivity [113]. This was attributed to the restoration of underrepresented bacterial groups and bacterial metabolites in the patients. Insulin resistance has been associated with high levels of serum branched-chain amino acids [114,115] and an increased population of gut microbes such as *Prevotella copri* and *Bacteroides vulgatus*, which are known branched-chain amino acid producers [116]. The gut microbiota of T2DM patients has also been characterized by reduced butyrate-producing bacteria and high levels of opportunistic pathogens [117]. Both obese and diabetic patients have high levels of members of the Lachnospiraceae family [118], *Enterobacter* and *Escherichia* [119]. On the other hand, an increase in the abundance of *Akkermansia muciniphila* (mucin-degrading gut bacteria) has been associated with metabolic health in obese [120] and diabetic patients [121]. Supplementation with *A. muciniphila* in obese and overweight human volunteers improved insulin sensitivity, reduced insulinemia, decreased total cholesterol, and reduced body weight. Recipients of the bacteria also showed improved liver functions and reduced systemic inflammation [120,122]. The health effects of *A. muciniphila* have been partly attributed to the presence of a heat-stable outer membrane protein Amuc_1100 [123] and pili-like protein MucT [81], which interact with Toll-like receptor 2 and Toll-like receptor 4 that improved gut barrier functions.

### 5.2. Autoimmunity

One of the possible causes of autoimmune damage of cells is molecular mimicry [124]. This could occur when normal commensal microbes mimic human antigens leading to reprogramming of the immune system and subsequent damage of host tissues [46]. Autoimmune damage could be triggered by microbial cell components such as lipopolysaccharide (LPS) or microbial metabolites that result in severe pathological conditions [46]. This was demonstrated in a European study involving infants with high levels of *Bacteroides* species and those with low levels of the same bacteria. The infants with high levels of *Bacteroides* spp., particularly *Bacteroides dorei*, were exposed to high levels of *Bacteroides* LPS, which triggered an autoimmune response leading to type-1 diabetes mellitus (T1DM). However, this was not recorded in babies with low levels of *Bacteroides* spp. [125]. Other studies have also shown that children with T1DM have reduced levels of podoviruses compared to those of children without the disease [126]. All these agree with earlier studies that suggested that gut microbial alterations preceded T1DM [127]. In several diseases including rheumatoid arthritis, T1DM, ulcerative colitis, systemic lupus erythematosus, and anti-phospholipid syndrome, autoantibodies against *Saccharomyces cerevisiae* cell wall component, phosphopeptidomannan, have been detected at high levels in patients [128]. In rheumatoid arthritis (RA) (a chronic autoimmune disease), patients experience inflammation and joint pain with various degrees of systemic involvement [129]. The guts of RA patients have been shown to be enriched with the proinflammatory bacteria group *Collinsella*, which can cause gut-membrane permeability [130]. The patients are also known to possess high levels of *Prevotella copri* in their gut [131,132,133] and large amounts of *N*-acetylglucosamine-6-sulfatase as well as filamin A (autoantigens associated with *Prevotella* peptides) in their synovial fluids [134]. Although available evidence implicates the gut microbiota in disease pathogenesis, further studies are needed to identify specific microbial strains and metabolites that are responsible for these phenotypes.

### 5.3. Inflammatory Bowel Disease (IBD)

Many studies have found a strong association between inflammatory bowel disease (Crohn’s disease (CD) and ulcerative colitis (UC)) and dysbiosis of the gut microbiota [20,135,136,137]. Regarding the association between gut mycobiota and IBD, disease severity was found to be correlated with fungal representations [138], and some studies have reported significant increase in the levels of Basidiomycota and reduced levels of Ascomycota [20]. They also have decreased populations of *S. cerevisiae* but increased levels of *Candida albicans* relative to healthy subjects [20]. The gut virome of IBD patients also displays an overrepresentation of Retroviridae family of viruses [139] and Caudovirales bacteriophages relative to healthy controls, though the increase was disease and cohort specific [140]. Similarly, IBD is characterized by loss of gut bacterial diversity [141] and enrichment of specific bacterial clades such as the *Ruminococcus gnavus* clade [142] and Enterobacteriaceae [143,144]. Recent studies have described the mechanism by which adherent–invasive *Escherichia coli* (AIEC), an opportunistic pathogen in the gut could play a crucial role in IBD [145]. Briefly, AIEC can evade the host immune system and attach to the gut epithelial cells while suppressing autophagy in IBD patients. The bacterium is translocated to the lamina propria, engulfed by macrophages, which results in the secretion of high amounts of TNF without causing host cell death. This leads to gut inflammation and AIEC over-colonization [137]. A meta-analysis and systematic literature review revealed that *Faecalibacterium prausnitzii* is severely reduced in IBD relative to healthy controls [146], which implies that supplementation with the bacterium could reduce inflammation associated with the disease [147]. A recent study has also shown that *Blautia, Faecalibacterium,* and *Ruminococcus* species could be keystone taxa in CD and UC [148]. Though these keystone taxa exert their influence on microbiome functioning irrespective of abundance, their presence may not necessarily guarantee their influence [149].

One possible mechanism by which *F. prausnitzii* reduces inflammation is its ability to produce a 15 kDa anti-inflammatory protein that inhibits the NF-κB pathway in gut epithelial cells [150,151]. Gut microbial metabolites of IBD patients could stimulate human dendritic cells to increase the ratio of Th2 to Th1 cells depending on the degree of disease severity [96]. The gut of IBD patients has reduced concentrations of tryptophan-derived indole derivatives that promote IL-22 production due to dysbiosis [152]. To confirm the association between IL-22 levels and IBD, fecal samples from *Card9*^−/−^ mice were transferred to wild-type germ-free mice and the recipients became susceptible to colitis. However, administration of tryptophan-metabolizing *Lactobacillus* strains restored intestinal IL-22 production and mitigated gut inflammation [152]. Knowing that *Card9*-signaling stimulates the production of IL-1β in the damaged bowel and controls the subsequent generation of IL-22 by group3 innate lymphoid cells [153], the ability of lactobacilli to attenuate colitis indicates that the gut microbiota and their metabolites are involved in bowel inflammations.

### 5.4. Atopic Asthma

The cause of asthma in children has not only been attributed to genetic risk factors but also to altered environmental exposures [154]. Most of the exposures, including the mode of delivery [32,155,156], term or preterm [157], drug intake [158,159,160], diet [161,162], pets and environmental toxins [163] are found to impact on the gut microbiome. The similarity in morphology and function of gut-associated lymphoid tissues and inducible bronchus-associated lymphoid tissues suggest a connection between the gut and lungs [164]. In fact, many lung diseases can be influenced by changes in the gut microenvironment and vice versa, and there are many lines of evidence that the gut microbiota is the link between the two sites [46,165]. The gut–lung axis is bidirectional, as it has been shown that stimulating the lungs with lipopolysaccharide could significantly alter the gut bacteria [166] and that pneumonia induces gut injury and retards gut epithelial proliferation [167]. Studies of geographically distinct babies from different socioeconomic and racial groups frequently report a depletion of normal commensal bacterial groups, including *Akkermansia*, *Faecalibacterium*, and *Lachnospira* in babies at risk of developing asthma or atopy [168,169,170]. Specifically, Arrieta et al. [168] observed that although atopic wheeze was associated with gut dysbiosis, the dysbiosis in rural Ecuadorean babies involved different bacterial and fungal taxa than that in Canadian babies who had the same disease. Several other studies have shown how environment and the gut microbiota are associated with the disease [171,172]. Current evidence suggests that gut dysbiosis precedes asthma and atopy in children and that low gut microbial diversity within the first month of birth is strongly associated with asthma in later life [173]. In a study of 298 children (aged 1–11 months), babies who had a lower relative abundance of *Bifidobacterium, Akkermansia*, and *Faecalibacterium* and a higher relative abundance of *Candida* and *Rhodotorula* were more susceptible to atopy and asthma. Their fecal samples contained 12,13-DiHOME (a gut microbial metabolite), which stimulated CD4^+^ cells to overproduce IL-4 while reducing the levels of CD4^+^CD25^+^FOXP3^+^ cells. Taken together, the study showed that gut dysbiosis could promote CD4^+^ T cell dysfunction, which is typically associated with childhood atopy [174]. Metagenomic sequencing of neonatal fecal bacteria showed an enrichment of bacterial epoxide hydrolase genes in the microbiome of neonates who developed asthma or atopy during childhood [175]. When expressed, epoxide hydrolase enhances the production of 12,13-DiHOME, which promotes allergic inflammation in the gut [172]. However, short-chain fatty acids (SCFAs) produced from high-fiber metabolism by gut bacteria could significantly reduce allergic airway inflammation [176]. Other similar studies showed that microbial butyrate, acetate, and propionate decreased T cell and dendritic cell activities and reduced the levels of IL-4 and circulating IgE for ameliorating airway inflammation [177]. Current available data indicate that the gut microbiome can be manipulated and their metabolites could influence inflammation in the lungs and bone marrow–derived immune cells [178]. This shows that the gut microbiota of early life contributes immensely to the risk of childhood asthma onset, and this indicates that the neonatal stage is the right time for implementing gut microbial interventions in high-risk babies.

### 5.5. Autism Spectrum Disorder

The gut microbiota has been recognized as an important mediator of signal transduction between the gut and the central nervous system [179]. This is partly because many neurodevelopmental disorders, including autism spectrum disorder (ASD), are strongly associated with gastrointestinal disorders [180,181]. Children with ASD have severely altered gut microbiota exhibiting increased levels of *Alistipes*, *Bilophila*, *Dialister*, *Parabacteroides*, *Veillonella*, as well as an enrichment of *Collinsella*, *Corynebacterium*, and *Lactobacillus* relative to control cohorts [182]. An ASD-associated gut microbiota also has significantly higher levels of fungal pathobiont *Candida* relative to the control group [182], with severely altered gut microbiota–associated epitopes [183]. Recent studies in a Chinese cohort have also ascertained the correlation between maternal microbiota and ASD in children. In their work, Li et al. [184] observed that mothers of ASD children had a high relative abundance of Proteobacteria, Alphaproteobacteria, Moraxellaceae, *Acinetobacter,* and a significant reduction in *Faecalibacterium* than mothers of healthy children. Moreover, both children with ASD and their mothers had high levels of *Clostridium* and *Streptococcus* [184]. Bacteria such as *Clostridium*, *Streptococcus*, *Chryseobacterium*, *Haemophilus*, and *Comamonas* are known to be involved in gastrointestinal disorders, maternofetal immune activation, maternal inflammation, and neonatal sepsis [184,185,186]. Since the gut microbiota development of neonates has been associated with maternal health [187] and maternal gut microbiota [184,188], it is possible that maternal immune activation [189,190] during pregnancy could play a role in the development. Gut dysbiosis in ASD results in significantly high concentrations of trimethylamine (TMA) [191], ammonia-N-oxide [192], metabolites that become neurotoxic when in high concentrations [193], and low levels of 2-keto-glutaramic acid [194] in the gut. Generally, transferring microbiota from healthy donors to ASD cohorts has been shown to result in significant improvement in gastrointestinal symptoms of diarrhea, constipation, abdominal pain, and indigestion while improving behavioral symptoms [180,195]. The recipients experience improvement in gut bacterial diversity and significant increase in the relative abundance of *Prevotella*, *Bifidobacterium*, and *Desulfovibrio* [180,195]. In other studies, administration of a probiotic cocktail of *Lactobacillus acidophilus, Lactobacillus rhamnosus*, and *Bifidobacterium longum* (10^6^ CFU) once daily for three months significantly improved gastrointestinal symptoms and autism symptoms compared to baseline [196]. Similarly, feeding children who have ASD with *Lactobacillus plantarum* WCFS1 (10^10^ CFU) for 12 weeks improved ASD symptoms and gastrointestinal conditions relative to baseline [197]. Furthermore, feeding ASD children with prebiotic galactooligosaccharide for six weeks significantly increased the relative abundance of *Faecalibacterium prausnitzii* and *Bacteroides* spp., while improving bowel movement and reducing abdominal pain [198]. All these findings support the role of the gut microbiota in modulating neuro-behavior and give a clue about the possibility of preventing and managing/treating behavioral disorders. Meanwhile, more studies are still needed to understand the mechanisms underlying the ability of gut microbial restoration to mitigate behavioral disorders.

## 6. The Microbiome as a Therapeutic Target

### 6.1. Dietary Interventions

Many studies have already shown the ability of diet to alter the gut microbial composition and functions, but only a limited number of controlled, clinical intervention studies aimed at modulating human gut microbes have been reported [1]. A recent study has shown that consumption of diets low in fermentable oligosaccharides, disaccharides, monosaccharides, and polyols could significantly increase the relative abundance of *Akkermansia*
*muciniphila* and butyrate-producing *Clostridium* cluster XIVa, while reducing the levels of *Ruminococcus torques* in patients with Crohn’s disease [199]. Feeding T2DM patients with a high-fiber diet effectively promoted the relative abundance of SCFA producers in the gut and improved glucose metabolism [200]. However, it has been shown that postprandial glycemic responses after feeding can be significantly variable because they are individual dependent [201]. A number of factors including gut microbial composition as well as fasting blood glucose levels and body mass indices appear to affect glucose response predictions, thus calling for the design of personalized diets for better glycemic control [202]. In another study, feeding obese women with flaxseed mucilage significantly altered the gut microbiota and improved insulin sensitivity [203]. Other researchers have also studied the impact of dietary nucleotides on infant gut microbiota. Specifically, Singhal et al. [204] observed that dietary nucleic acid supplemented infant formula could significantly modify the gut bacteria of infants. Although it is evident that diet influences gut microbes, future studies must consider nutritional strategies tailored toward the growth or reduction of specific microbial communities during dysbiosis and disease. Such a microbiome-targeted approach based on independent studies in gut microbiome research may offer more accurate, predictable, and sustainable microbial restoration in pathological conditions caused by gut microbiome dysbiosis.

### 6.2. Multispecies Microbial Supplements

Although the use of probiotics to modulate the gut microbiota has shown promising results in animal studies, results from human trials are still inconclusive. Recent studies have shown that probiotics display cohort-, region-, and strain-specific mucosal colonization patterns determined by predictive baseline host and microbiome features [205,206]. Probiotic supplementation may thus be limited in their universal and persistent impact on the gut mucosa, thus calling for the development of personalized probiotics. Single formulations and cocktails of probiotic bacteria and fungi have been administered to humans, and promising results have been recorded. For instance, in a randomized control trial, children with peanut allergies who received either placebo or oral peanut immunotherapy combined with *Lactobacillus rhamnosus* GG (once daily for 18 months) remained desensitized to peanuts even after 4 years of the intervention relative to infants in the placebo group [207]. Supplementing casein formula with *L. rhamnosus* GG and feeding it to children with cow’s milk allergy promoted tolerance, increased butyrate production, and influenced their gut bacterial community profile relative to control groups [208]. A recent meta-analysis has shown that *Lactobacillus rhamnosus* GG may effectively prevent antibiotic-associated diarrhea in infants as well as adults treated with antibiotics [209]. However, *L. rhamnosus* GG colonization and its health benefits may be short lived [210], thereby suggesting the need for long-term or earlier interventions or the use of a different microbial consortia that could competitively colonize the newborn gut for sustained benefits [46]. Meanwhile, administration of a probiotic bacteria cocktail (*Lactobacillus acidophilus, Lactobacillus delbrueckii* subsp. *bulgaricus, Lactobacillus casei*, and *Lactobacillus plantarum*) effectively enhanced remission in patients with UC [211]. In another study, the same bacterial consortium also significantly improved fatty liver disease severity, reduced body mass index, and increased glucagon-like peptide levels in obese children [212]. Furthermore, supplementing probiotic to children at high genetic risk for T1DM during the first 27 days of life significantly reduced risk of islet autoimmunity compared to control groups [213]. Current studies are seeking to unravel the basis of microbe-host and microbe-microbe interactions so as to identify gut microbial metabolites, gut microbe-host co-metabolites, and specific microbes that play critical roles in health and disease. There is also the need to develop tailored probiotics for specific individuals since not all the population may benefit from a given probiotic despite its health claims. Such a strategy could result in the development of personalized multispecies bacterial consortia derived from a healthy human gut.

### 6.3. Phage Therapy

Bacteriophages are viruses that infect bacteria. They interact with bacteria through recognition of specific recognition proteins on bacterial cell surfaces. Recent studies have shown that a phage usually infects only a limited number of bacteria of the same species [214] and may impose significant bactericidal pressure on them [215]. However, some phages such as lytic phage Stau2 [216] have a broad spectrum. Thus, phages could be better tools for combating specific pathogenic enteric bacteria than antibiotics [217,218,219]. For this reason, current phage research in microbiome studies has focused on using phages to control gut pathobionts. A study in Poland showed that remission of bacterial infections in patients undergoing phage therapy could be as high as 90% [220]. Meanwhile, not all patients who underwent phage therapy recovered, and this may be due to the severity of their underlying injury [217]. An earlier study showed that administration of a single bacteriophage to an infant rabbit 1 h before a *Vibrio cholerae* challenge could protect the rabbit from cholera [221]. A more recent study has similarly reported that administration of a cocktail of three phages to healthy mice and rabbits effectively prevented *V. cholerae* infection [222]. Intraperitoneal injection of phi MR11 of a *Staphylococcus aureus* phage suppressed *S. aureus*-induced lethality after an *S. aureus* challenge in mice [223]. Meanwhile, though a consortium of five lytic bacteriophage types administered to adult rabbits either 6 or 12 h before a *V. cholerae* challenge only slightly decreased diarrheal severity; the treatment significantly reduced *V. cholerae* load and diarrheal severity when the phage was administered after the bacteria challenge [224]. This implies that some phages may work better as prophylactics while others function better when used as treatments. Phages have also been shown to protect immunocompromised animal models against bacterial infections and lethality [225,226]. For instance, intravenous or intraperitoneal administration of phage ΦSA012 (*S. aureus* phages) to a mouse mastitis model significantly reduced *S. aureus* proliferation and inflammation in the mammary gland [225]. A lytic phage strain (GRCS) also protected diabetic mice from lethal bacteremia after an *S. aureus* challenge [226]. In another study, antibiotics and phages were combined to synergistically clear *Pseudomonas aeruginosa* infection in animal models with endocarditis [227]. In human studies, intravenous and percutaneous administration of a consortium of nine different lytic bacteriophages into the abscess cavities of a diabetic patient with necrotizing pancreatitis significantly cleared the *A**cinetobacter*
*baumannii* infection and caused remission [228]. In a randomized, controlled double-blind phase ½ trial, administration of a combination of lytic anti-*Pseudomonas aeruginosa* bacteriophages decreased bacterial burdens in patients with burns infected by *Pseudomonas aeruginosa* [229]. Although phage therapy may show good outcomes, they may not offer sustained protection if the host is re-infected by the same bacterium after remission. This is because some phages can be deactivated or neutralized by human serum [230], and the rate of deactivation may depend on the route of phage administration. The phages may also be passed out through feces [231] and, thus, would reduce the therapeutic phage titers available in the gut.

## 7. Challenges and Perspectives

Metagenomics and metabolomics have made it possible to analyze gut microbe-microbe and microbe-host interactions, and the Human Microbiome Project (HMP) has been so far the biggest effort to characterize human microbial flora. Although in many diseases it is puzzling whether gut dysbiosis precedes the disease or dysbiosis is a result of the disease condition, metagenomics and metabolomics have provided strong evidence that gut microbial dysbiosis is involved in various human diseases that occur in the gut or at distant organs.

In spite of the fact that gut dysbiosis may serve as a prognostic biomarker of a disease as it may occur many years before the actual disease is manifested, it could be difficult to identify which specific disease the subject is at risk of. Furthermore, research has still not unveiled which particular bacterial groups or species specifically define a healthy gut or a given pathological condition. Considering that the gut microbiome of each individual is different, it may be helpful to analyze the gut microbiota of healthy individuals and store samples of their feces (personalized stool banking) for comparison from time to time. Such an approach will be helpful to easily detect dysbiosis and determine when to provide the necessary interventions. Such an approach will however have to take into consideration factors such as age, diet, and environmental conditions since they play critical roles in modifying the gut microbes. Meanwhile, recent efforts in the field of microbiome studies have unveiled specific mechanisms of the pathogenesis of certain diseases such as *Clostridioides difficile* infection, and this has enhanced new microbial targets for therapeutic development. Positive results have been reported for some disease conditions after patients received dietary interventions, probiotics, or a consortium of gut bacteria isolated from healthy people. The success of these strategies proves that the gut microbiome is an important therapeutic target, calling for the development of more personalized strategies for disease prevention, management, and treatment. Despite the promise, large and longitudinal integrative research works are needed in humans for gut bacteria, mycobiome, and virome including their products, which have immunological and physiological impacts. Accompanying such studies with mechanistic studies would be important to understand how microbe-host and microbe-microbe interactions impact human health and disease.

However, a key challenge in microbiome research is the technology used for analysis. In fact, most studies depend on amplifying rRNA genes to determine the microbial composition of samples. These studies are biased since different scientists may use different primers when carrying out PCR amplification. Moreover, different researchers tend to use different bioinformatics programs, which yield different outcomes. Since not all gut microbes have been identified, certain important microbes may not even be captured during analysis. However, if captured, their identities remain unknown because they may not be present in available databases. Hence, one cannot compare independent studies conducted by different scientists. This makes it difficult for scientists to make conclusive decisions regarding gut microbiota in patients with different disease states when the methods are so inaccurate and imprecise.

## Figures and Tables

**Figure 1 high-throughput-09-00017-f001:**
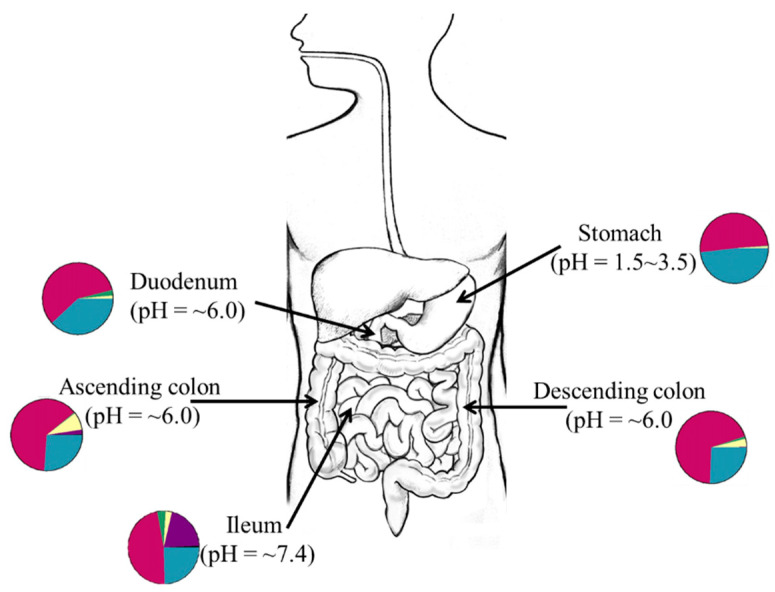
Graphical representation of the niche specificity of gut microbiota. Each gut compartment has a different pH, nutrient availability, and mucus structure. These factors may influence the microbial structure at any given time and space. The colors of the pie charts are arbitrary representations of the different microbial groups at different sites of the gut.

**Figure 2 high-throughput-09-00017-f002:**
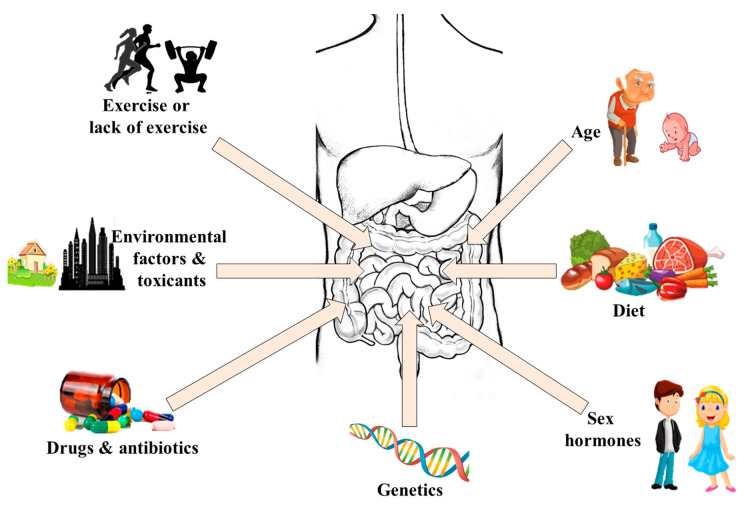
Factors that cause variations in the gut microbiota.

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
