# Peer review of "Health Impact and Therapeutic Manipulation of the Gut Microbiome"

_2571-5135, 2020, doi:10.3390/ht9030017_

Round 1

Reviewer 1 Report

In general, the paper has a logical progression through out and the subject area is very interesting, with many new aspects only revealed in the last decade or so. The title is quite wide and there is a risk that some crucial aspects are easily ignored. This manuscript suffers of it too, as there is nothing directly related to biofilms which are essential to some gut functions and health. Enterococci are one of the most abundant bacterial species in gut and they are linked to opportunistic infections, aided by biofilm formation. However, this aspect has not been covered in this review at all, even the word “biofilm” has not been mentioned in the text.

I would suggest rephrasing the title a little bit. For example, “Health impact and therapeutic manipulation of the gut microbiome” would be more understandable.

The English language and the used of different fonts, extra spaces italics, bold and underline should be checked for the whole manuscript, as at present the presentation of bacterial scientific names varies wildly throughout. For example, see line 263, 365 and 393.

Specific comments:

Line 23-24: The most recent study -> “A recent study”. This paper is from 2016 so nearly 5 years old.

Line 29: It should be pointed out, that this is the case from very early on in human life. Thus, the excellent review by Milani et al should be cited here, perhaps with a generic statement regarding this aspect. “The First Microbial Colonizers of the Human Gut: Composition, Activities, and Health Implications of the Infant Gut Microbiota. Milani C, Duranti S, Bottacini F, Casey E, Turroni F, Mahony J, Belzer C, Delgado Palacio S, Arboleya Montes S, Mancabelli L, Lugli GA, Rodriguez JM, Bode L, de Vos W, Gueimonde M, Margolles A, van Sinderen D, Ventura M. Microbiol Mol Biol Rev. 2017 Nov 8;81(4)"

Line 33: As cancer has been mentioned here, it would be logical to add any evidene of gut microbiota to colon cancer: Zackular JP, Baxter NT, Iverson KD, Sadler WD, Petrosino JF, Chen GY, Schloss PD. The gut microbiome modulates colon tumorigenesis. MBio 4(6):e00692-00613, 20

Also, one of the landmark papers regarding IBD and bacteria should be added here as well: Jostins L., Ripke S., Weersma R.K., Duerr R.H., McGovern D.P., Hui K.Y., Lee J.C., Schumm L.P., Sharma Y., Anderson C.A., et al. Host-microbe interactions have shaped the genetic architecture of inflammatory bowel disease. Nature. 2012;491:119–124

Line 45: As the subject has already been covered at least partially, it might be best to rephrase “There is evidence -> “There is more evidence…” and quote some relevant papers(s).

Line 58: Quorum sensing is such an important aspect of the microbe action in gut that it should be briefly described. Lines 58-63 need rephrasing as the use of English is not very clear here (tautology should be removed).

Line 68: No explanation for the pie charts. If this is based on the authors own data, this information could be cited at least some level, even if it is not published. If this is adapted from another paper, the source should be mentioned.

Line 76-86: The message should be clearer here. At present the text goes back and forth with monozygotic twins. It should be clearly said that even the twins have different microbiota if they are living in different environment/household, whereas family members living in the same household/environment have similar microbiota, even if they are not related. If this is the case, this could be concluded much more clearly here.

Line 95 (or so): It should be mentioned that initially the gut was considered pretty much sterile. The process to change this view started fairly late, less than 15 years ago with the human microbiome project.

Turnbaugh PJ, Ley RE, Hamady M, Fraser-Liggett CM, Knight R, Gordon JI. The human microbiome project. Nature 449(7164):804-810, 2007.

Line 118: “Very little works” need rephrasing. There are actually a couple of  high quality papers focusing on the virome or mycobiome and human immunity, or at least typical inflammation markers:

Kuss SK, et al. Intestinal microbiota promote enteric virus replication and systemic pathogenesis. Science. 2011;334:249–252.

Kane M, et al. Successful transmission of a retrovirus depends on the commensal microbiota. Science. 2011;334:245–249

Li, Q. , Wang, C. , Tang, C. , He, Q. , Li, N. , & Li, J. (2014). Dysbiosis of gut fungal microbiota is associated with mucosal inflammation in Crohn's disease. Journal of Clinical Gastroenterology, 48(6), 513–523

Ott, S. J. , Kuhbacher, T. , Musfeldt, M. , Rosenstiel, P. , Hellmig, S. , Rehman, A. , … Schreiber, S. (2008). Fungi and inflammatory bowel diseases: Alterations of composition and diversity. Scandinavian Journal of Gastroenterology, 43(7), 831–841

Hoarau, G. , Mukherjee, P. K. , Gower‐Rousseau, C. , Hager, C. , Chandra, J. , Retuerto, M. A. , … Ghannoum, M. A. (2016). Bacteriome and mycobiome interactions underscore microbial dysbiosis in familial Crohn's disease. mBio, 7(5), e01250‐16

Line 129 onwards: Our hunger sensing and obesity has been firmly associated to leptin and ghrelin peptides, and the gut microbiota has been linked to them as well as to intestinal signalling for fats, for example. Hence, these two papers should be cited, especially as at present this manuscript does not even mention these molecules.

Schéle E, Grahnemo L, Anesten F, Halleń A, Backhed F, Jansson JO. The gut microbiota reduces leptin sensitivity and the expression of the obesity-suppressing neuropeptides proglucagon (Gcg) and brain-derived neurotrophic factor (Bdnf) in the central nervous system. Endocrinology. 2013;154:3643–365

Duca FA, Swartz TD, Sakar Y, Covasa M. Increased oral detection, but decreased intestinal signaling for fats in mice lacking gut microbiota. PLOS ONE. 2012;7:e3974

Also, the circadian rhythms have been shown to affect BMI. This paper suggest a role for microbiota in this process as well.

Thaiss CA, Levy M, Korem T, Dohnalová L, Shapiro H, Jaitin DA, David E, Winter DR, Gury-Benari M, Tatirovsky E, Tuganbaev T, Federici S, Zmora N, Zeevi D, Dori-Bachash M, Pevsner-Fischer M, Kartvelishvily E, Brandis A, Harmelin A, Shibolet O, et al. Microbiota diurnal rhythmicity programs host transcriptome oscillations. Cell 167(6):1495-1510.e1412, 2016.

Line 138 and 162: Italics used in main text for unknown reason

Line 180-181: One of the landmark papers for Crohn’s disease should be added:

Seksik P., Rigottier-Gois L., Gramet G., Sutren M., Pochart P., Marteau P., Jian R., Dore J. Alterations of the dominant faecal bacterial groups in patients with Crohn’s disease of the colon. Gut. 2003;52:237–242. This paper indicates biodiversity of the microflora and the role of Enterobacteria in CD

Line 184: References required. These should be covered as well:

Srivastava A, Gupta J, Kumar S, Kumar A.Gut biofilm forming bacteria in inflammatory bowel disease. Microb Pathog. 2017 Nov;112:5-14.

Watnick P, Kolter R. Biofilm, city of microbes. J Bacteriol. 2000 May;182(10):2675-

Line 196: Severely reduced compared to what, a control group?

Line 198: This review would be beneficial for the reader:

Banerjee S, Schlaeppi K, van der Heijden MGA. Keystone taxa as drivers of microbiome structure and functioning.Nat Rev Microbiol. 2018 Sep;16(9):567-576.

Line 205: Worth mentioning this paper as it explains bit more of the relations hip between Card9 and Il-22:  Bergmann H, Roth S, Pechloff K, Kiss EA, Kuhn S, Heikenwälder M, Diefenbach A, Greten FR, Ruland J.Card9-dependent IL-1β regulates IL-22 production from group 3 innate lymphoid cells and promotes colitis-associated cancer. Eur J Immunol. 2017 Aug;47(8):1342-1353

Line 217-220: This sentence should be either split in two or rephrased for clarity.

Line 220: The sentence starting “However,…” has very a low information content and could be removed.

Line 228: This sentence has italics again and the use of English should be revised.

Line 232: “In a study in…” > “in a study of….

Line 239-241: Sentence should be rephrased. Not clear if the genes are present or are they expressed and the end of line 240 does not make sense.

Line 253: Many studies have demonstrated that the gut microbiota plays a critical role in the metabolism of serotonin. Serotonin (5-HT) has been shown to have a pleiotropic function in gastrointestinal, neurological/psychiatric and liver diseases. Also the peripheral dopamine is controlled by gut microbes and have been shown to inhibit T-cell mediated hepatitis. These aspects could be covered at least briefly.

Line 291: Not sure if [166] is the appropriate reference here.

Line 303: One of the most common supplements containing supplements is yeast extract. However, it has mostly purines but has been linked to reduced intestinal inflammation in pigs.

Animal. 2017 Dec;11(12):2156-2164. Dietary supplementation with a nucleotide-rich yeast extract modulates gut immune response and microflora in weaned pigs in response to a sanitary challenge.Waititu SM1, Yin F1, Patterson R2, Yitbarek A1, Rodriguez-Lecompte JC3, Nyachoti CM1.

J Anim Sci. 2019 Dec 17;97(12):4875-4882. Supplemental effects of dietary nucleotides on intestinal health and growth performance of newly weaned pigs. Jang KB1, Kim SW.

Line 340-376 : This section is pestered with font problems, possibly written in a hurry?

Line 377-402: In general, this paragraph needs English language revision. Sentences in lines 379 and 383 start with “Although…”, sentences in lines 289 and 290 start with “Such an approach…”. The concept of personal stool banking has not been explained further, for example from the transplantation point of view.

Reviewer 2 Report

High throughput review: 768062

Overview: This review surveys studies that investigated the relationship between the gut microbiome and various disease states: obesity and type 2 diabetes mellitus, autoimmunity, inflammatory bowel disease, atopic asthma, and autism spectrum disorder. The authors discuss the microbiome as a therapeutic target that can possibly be manipulated by dietary interventions, microbial supplements, and phage therapy. The authors state that there is not enough evidence on the role of gut metabolites and the mechanisms by which the microbiota affect health and disease. The real question to be addressed in coming years is whether gut dysbiosis precedes pathology or the other way around?

Strength: The topic is relevant and the question presented needs to be addressed. The review is timely.

Weakness: Organization, digestibility, need to address the question, and no mention of how DNA sequencing technology and other methods might be a source of the problem.

Problems:

  • The readers need to know explicitly what the authors are going to talk about. For example, Sections 2 to 4. Although important sections, it is not obvious in the introduction why the authors need to mention niche specificity of the gut microbiota, the variability of the gut microbiome, and the role of the microbiota in immunity and homeostasis. This could be easily resolved by putting a statement in the introduction that ties the sections all together.
  • Reader digestibility. Each paragraph contains a lot of interesting information from the literature, which at times, I found overwhelming. It might help to make the article more digestible by making tables that summarize the presented information with references.
  • The question presented is very interesting: whether gut dysbiosis precedes pathology or the other way around? I suggest that the author more explicitly address the question in the last section. The section is “fuzzy”.
  • Technology is a big factor in the studies mentioned – yet the authors never discuss it. For example, most studies depend on amplifying rRNA genes to determine the microbial composition. These studies are biased due to the fact they are based on PCR amplification with different primers. Moreover, different bioinformatics programs will yield different results. Hence, one cannot compare studies. One can argue that it is difficult for scientists to say anything about gut microbiota in patients with different disease states when the methods are so inaccurate and imprecise.

Minor points:

Italic problems and irregular font changes

Line 109, 138, 147,152, 153, 163, 228, 256, 257, 262, 293, 365

Line 285 “mores” ?

Round 2

Reviewer 1 Report

Line 61: ”While others (relatively recent studies) propose”. Should be ”...white other relatively recent studies” or ”...while others propose”. Since ”many early studies” and ”relatively recent” studies mentioned here are all from 2015-2017, I would suggest to go without ”recent” mentioned.

Line 79:. In the author response, the authors write ”The figure is the authors own drawing. We have indicated that the pie charts are arbitrary representations of gut microbial populations”. Actually this is not true. The figure is clearly partially copied from reference 29: ”Repertoire of the Gut Microbiota From Stomach to Colon Using Culturomics and Next-Generation Sequencing”. In text line 57, the reader gets an impression that the Figure 1 is based on papers 28-30. However, there is nothing in the figure legend regarding this. The partial plagiarism issue and credits to original source needs to be sorted before the acceptance of the manuscript can be considered. Options are for example using phrasing like ”modified from” or ”based on”, with the source paper given.

Line 93: The central illustration is property of PIXOLOGICSTUDIO/Getty Images. The editor-in-chief should verify with the authors that there are no copyright issues, as those can turn out to be  expensive.

https://www.verywellhealth.com/picture-your-digestive-system-1945313?utm_source=pinterest

Line 134: relatively few works -> relatively little work

Line 282: ”also have extremely higher levels” ->

Line 431-440: This new addition should look back but also have some future prospects. The text inserted applies for NGS but this has not even been mentioned. Also, the Human Microbiome Project has been so far the biggest effort to characterise the human microbial flora. This project ran from 2007-2016 and received nearly $200 million funding so it should be mentioned here, as at present this section gives the impression that only ”fishing expeditions” by individual scientists have been performed. The current text here can be shortened, for example by using PCR abbreviation instead of the full name.
